# The Effects of Prophylactic Laxative Use on Critically Ill Patients Requiring Mechanical Ventilation: A Retrospective Cohort Study

**DOI:** 10.3390/diseases12110274

**Published:** 2024-11-01

**Authors:** Heqing Tao, Ligang Liu, Weipeng Lu, Ziyan Ni, Xueqing Chen, Milap C. Nahata, Liang Peng

**Affiliations:** 1Department of Gastroenterology, The First Affiliated Hospital of Guangzhou Medical University, Guangzhou 510120, China; tao_heqing@163.com (H.T.); luweipeng2018@163.com (W.L.); ni_zi_yan@outlook.com (Z.N.); chenxq@vip.163.com (X.C.); 2Institute of Therapeutic Innovations and Outcomes, College of Pharmacy, The Ohio State University, Columbus, OH 43210, USA; liu.10645@osu.edu; 3College of Medicine, The Ohio State University, Columbus, OH 43210, USA

**Keywords:** laxatives, mechanical ventilation, critical care, constipation, mortality

## Abstract

**Objective**: To investigate the effects of prophylactic use of stimulant laxatives and/or docusate on the clinical outcomes in critically ill patients who required mechanical ventilation (MV). **Methods**: A single-center, retrospective, cohort study was conducted. Patients who received MV in the first 24 h after intensive care unit (ICU) admission were enrolled and divided into four groups: non-laxative, stimulant laxatives, docusate, and stimulant laxatives–docusate combination. The primary outcome was in-hospital mortality. The major secondary outcomes included ICU-free days and ventilator-free days at 28 days; the other outcomes were ventilation-associated pneumonia (VAP), enterobacterial infection, diarrhea, and electrolyte disturbances. Inverse probability treatment weighting (IPTW) was used to adjust for confounders. **Results**: A total of 2129 patients were included in this study, 263 of whom received stimulant laxatives, 253 received docusate, 368 received a combination, and 1245 did not receive any laxative. The prophylactic use of docusate was associated with a decreased risk of in-hospital mortality (OR: 0.59, 95% CI 0.42 to 0.83, *p* = 0.002) and VAP (OR: 0.62, 95% CI 0.47 to 0.81, *p* = 0.001). It was also associated with an increase in ICU-free days at 28 days (β: 0.89, 95% CI 0.83 to 0.95, *p* < 0.001). Importantly, laxatives prophylaxis was not associated with increased risks of diarrhea, electrolyte disturbances, and enterobacterial infections. **Conclusions**: Prophylactic use of docusate may improve certain prognoses and does not demonstrate any adverse events. However, further research is necessary to determine the optimal regimen and dosage of prophylactic laxatives in this specific population.

## 1. Background

Mechanical ventilation (MV) is a common life-saving technique used in critical care settings. However, it can lead to constipation due to decreased intestinal motility, impaired visceral perfusion, and increased pro-inflammatory mediators [1]. A lack of bowel movements for three days or more is generally considered to be constipation [2,3,4,5]. It has been estimated that up to 15% of ventilated patients experience constipation [1].

Constipation has been associated with various adverse outcomes, such as a higher incidence of ventilator-associated pneumonia (VAP), prolonged hospitalization, and even a higher mortality rate [4,6]. Delivering bowel care to mechanically ventilated critically ill patients may be beneficial. Different organizations have recommended bowel protocols for managing constipation in ICUs, with the commonly used medications including senna, bisacodyl, sodium phosphate, glycerin, and docusate [7]. Docusate and stimulants are commonly used laxatives in the hospital setting, particularly in mechanically ventilated critically ill patients. However, none of the prior studies investigated the association between the use of stimulant laxatives or docusate as individual treatments, or their combination, on the clinical outcomes in ICU patients requiring mechanical ventilation. Therefore, we performed this retrospective observational study to investigate the potential effects of prophylactic use of stimulant laxatives and/or docusate on the clinical outcomes in mechanically ventilated critically ill patients.

## 2. Methods

### 2.1. Study Design and Participants

This was a single-center, retrospective, observational cohort study that adhered to the Strengthening the Reporting of Observational Studies in Epidemiology (STROBE) recommendations [8].

This study enrolled patients aged 18 years or older who required mechanical ventilation within the first 24 h after ICU admission and continued for more than 48 h within 72 h of ICU hospitalization. Individuals with ICU stays of less than 72 h, those who received laxatives other than stimulant laxatives and docusate during their ICU stay, those with a duration of prophylactic laxative use of less than three days, those with the use of opioid antagonists, or those already diagnosed with constipation with International Classification of Diseases, Ninth Revision (ICD-9) and Tenth Revision (ICD-10) diagnosis codes were excluded. Patients who had post-gastrointestinal surgery, gastrointestinal ostomy, and those with ileus were also excluded. If a patient had more than one ICU admission, only the first admission was included in this study.

### 2.2. Data Source

Data were extracted from a publicly accessible, online, open-source database, the Medical Information Mart for Intensive Care IV (MIMIC IV) database, which contained comprehensive information from medical records from 2008 through 2019 at the Beth Israel Deaconess Medical Center, a tertiary medical institution in Boston, USA. The Structured Query Language (SQL) was used to download data, including patient-specific information, medication history, procedure records, laboratory results, and mortality.

### 2.3. Prophylactic Laxatives Use Definition

Prophylactic use of laxatives was defined as the administration of stimulant laxatives and/or docusate during the first 24 h of the ICU stay to patients to prevent constipation. The stimulant laxatives included senna and bisacodyl. The entire cohort was divided into four groups: non-laxative, stimulant laxatives, docusate, and stimulant laxatives–docusate combination.

### 2.4. Outcome Definition

The primary outcome was in-hospital mortality. The major secondary outcomes included ICU-free days and ventilator-free days at 28 days; the other secondary end points consisted of the incidence of ventilation-associated pneumonia, enterobacterial infection, diarrhea, and electrolyte disturbances. Enterobacterial infections included Enterobacteriaceae detected in any specimen and Clostridioides difficile (C. difficile) from stool specimen. The specimens used to detect Enterobacteriaceae included blood and respiratory samples, including bronchial brush, bronchial washings, bronchoalveolar lavage, transtracheal aspirate, and sputum. Electrolyte disturbances included hypernatremia (serum sodium > 145 mmol/L), hypokalemia (serum potassium < 3.5 mmol/L), and hypomagnesemia (serum magnesium < 1.8 mg/dL). Enterobacterial infection and electrolyte disturbance were defined as at least a one-time occurrence of Enterobacterial infection and electrolyte disturbance within 48 h after admission to the ICU and 48 h after discharge from the ICU. Patients who defecated more than three times per day and had a score on the Bristol stool scale of more than 6 points were considered to have diarrhea.

### 2.5. Covariates Identified

We identified the following 30 covariates: patient characteristics (age, sex, weight), type of ICU (medical intensive care unit (MICU), surgical intensive care unit (SICU)/trauma surgical intensive care unit (TSICU), coronary care unit (CCU)/cardiac surgery recovery unit (CSRU)), SOFA scores, Acute Physiology Score III (APSIII), Charlson Comorbidity Index (CCI), comorbidities including diabetes, congestive heart failure, liver disease, renal disease, and chronic obstructive pulmonary disease (COPD), vasopressors and opioid administration, mean arterial pressure (MAP), complete blood counts, electrolytes, chemistries, and arterial blood gases (ABG). All the baseline laboratory test results, vasopressor use, and MAP were obtained within 24 h of admission. If there were multiple test results for the same item, we only kept the most severe one. All the comorbidities were identified by the International Classification of Diseases, Ninth Revision (ICD-9) and Tenth Revision (ICD-10) diagnosis codes (Appendix A).

### 2.6. Statistical Analysis

In the preliminary processing of raw data, variables with more than 10% missing values were excluded. The frequency of missing values for the variables is presented in Appendix A. Covariate measurement records that extraordinarily deviated from the respective reference intervals were replaced by mean substitution. Multiple imputation using the “mice package” in R was employed for data imputation in this study.

Univariate analyses were conducted to examine the association of different variables and each predefined outcome in this study. Variables that demonstrated statistical significance in the univariate analysis were subsequently included in the respective multivariate analysis models. Initially, the Cox proportional hazards regression model was intended for use in analyzing in-hospital mortality. However, upon examination of the Schoenfeld residuals, it became evident that the proportional hazards assumption was violated (Appendix A). Consequently, logistic regression was employed instead.

Multivariable logistic regression was used to examine the association between early use of laxatives and VAP, enterobacterial infections, and electrolyte abnormalities. The normality of ICU-free days and ventilator-free days at 28 days was assessed. For variables that demonstrated a normal distribution, multiple linear regression was utilized. Alternatively, generalized linear models with a Poisson or negative binomial distribution were applied for variables that did not conform to normality. As an alternative approach to ensure a balance among the four groups, we implemented inverse probability treatment weighting (IPTW). Subsequently, logistic regression was employed to analyze the association between laxative prophylaxis and outcomes except for ICU-free days and ventilator-free days at 28 days, which were analyzed via generalized linear regression. The variance inflation factor was used to determine multicollinearity among the variables in the regression model.

Subgroup analyses were performed for in-hospital mortality based on several variables, including age (<65 or ≥65 years), gender (male or female), type of ICU (MICU, SICU/TSICU, or CCU/CSRU), ASPIII (<40 or ≥40), CCI (3 or ≥3), SOFA score (<5 or ≥5), COPD (yes or no), diabetes (yes or no), renal disease (yes or no), and use of vasopressors (yes or no). Interaction tests were used to assess whether the relationship differed across different groups.

To examine the potential impact of unmeasured or residual confounding factors on in-hospital mortality, a sensitivity analysis was conducted using the E-value [9]. The E-value represented the minimum strength of association that an unmeasured confounder completely explained the observed treatment effect estimate. This analysis was performed to assess the robustness of our results and to evaluate the potential influence of unmeasured confounding on the observed treatment effects.

The baseline characteristics of the original and reorganized cohorts were compared, with the N (%) for categorical variables. Continuous variables with a normal distribution were presented as the mean (standard deviation), while continuous variables without a normal distribution were reported as the median (interquartile range). Standardized mean differences (SMDs) were calculated to compare the inter-group differences of the respective covariates. The covariates were considered to be effectively balanced if their SMD was less than 0.10. All the statistical analyses were conducted using R (version 4.1.3). To address the inflated type I error, a Bonferroni multiple testing correction was employed. In this correction, a significance level of <0.0125 was used for the primary analyses, while a significance level of <0.005 was used for the subgroup analyses. These adjusted thresholds were considered to indicate statistical significance.

## 3. Results

A total of 2129 patients were enrolled in the original cohort, of whom 263 received stimulant laxatives, 253 received docusate, 368 received stimulant laxatives and docusate combination, and 1245 did not receive any laxatives (Figure 1). The baseline characteristics of the original cohort are presented in Table 1. Covariates such as age, type of ICU, APSIII score, CCI, diabetes, renal disease, COPD, blood pH, blood urea nitrogen (BUN), vasopressor use, and enteral nutrition use showed statistical significance. Patients in the docusate group were older than those in the other groups (*p* < 0.001). For patients receiving docusate, the APSIII score was lower, and the SOFA score and CCI were higher, compared to patients in the other three groups.

The results of the univariate analysis of in-hospital mortality appear in Appendix A. The age, female gender, weight, APSIII score, SOFA score, CCI, renal disease, vasopressor usage, temperature, blood pH, pCO2, BUN, serum creatinine (SCr), and lactate were variables of statistical significance. These variables were selected as confounders in both the multivariate regression analysis and the IPTW analysis. After the IPTW adjustment, except for the SOFA score, all the covariates included in the regression model were adequately balanced, with their SMD being substantially less than 0.10 (Figure 2A). The variance inflation factor of all the covariates did not exceed three, meaning no obvious multicollinearity existed among these variables (Figure 2B).

### 3.1. Primary Outcome

In-hospital mortality was reported in 378 (30.4%) patients who did not receive laxative, 65 (24.7%) patients who received stimulant laxatives, 48 (19%) patients administered docusate, and 119 (32.3%) patients who received both stimulant laxatives and docusate. Prophylactic use of docusate was associated with decreased in-hospital mortality in the logistic regression (OR: 0.48, 95% CI 0.33 to 0.69, *p* < 0.001) and the IPTW model (OR: 0.59, 95% CI 0.42 to 0.83, *p* = 0.002). Prophylactic use of stimulant laxatives did not significantly affect in-hospital mortality with the logistic regression (OR: 0.86, 95% CI 0.76 to 1.41, *p* = 0.37) and the IPTW method (OR: 1.03, 95% CI 0.76 to 1.41, *p* = 0.834). A similar finding was obtained for the combination therapy in the logistic regression (OR: 1.23, 95% CI 0.94 to 1.61, *p* = 0.13) and the IPTW method (OR: 1.28, 95% CI 0.98 to 1.67, *p* = 0.069) (Table 2 and Appendix A).

### 3.2. Major Secondary Outcomes

#### 3.2.1. ICU-Free Days at 28 Days

Although docusate was not associated with an increase in ICU-free days at 28 days in the multivariate regression analysis (β: 0.92, 95% CI 0.85 to 0.99, *p* = 0.028), its use increased the ICU-free days at 28 days in the IPTW model (β: 0.89, 95% CI 0.83 to 0.95, *p* < 0.001). Stimulant laxatives and the combination did not significantly increase the ICU-free days at 28 days in both models (Table 2 and Appendix A).

#### 3.2.2. Ventilator-Free Days at 28 Days

In both the logistic regression and IPTW models, docusate alone, stimulant laxatives alone or the combination therapy demonstrated no statistically significant impact on the ventilator-free days at 28 days (Table 2 and Appendix A).

### 3.3. Other Secondary Outcomes

#### 3.3.1. Ventilation-Associated Pneumonia

Compared with the non-laxative users, the use of docusate decreased the risk of VAP in the IPTW model (OR: 0.62, 95% CI 0.47 to 0.81, *p* = 0.001). However, neither stimulant laxatives nor the combination showed a significant association with the risk of VAP in both the logistic regression analysis and the IPTW models (Table 2 and Appendix A).

#### 3.3.2. Enterobacterial Infections

Docusate, stimulant laxatives, and the combination of docusate and stimulant laxatives were not significantly associated with the risk of Enterobacterial infection and C. difficile infection in both the logistic regression analysis and IPTW analysis (Table 2 and Appendix A).

#### 3.3.3. Diarrhea

Compared with the non-laxative group, all three groups were not associated with an increased or decreased risk of diarrhea in both the logistic regression analysis and the IPTW model. However, docusate was associated with a trend toward a decreased risk of diarrhea (OR: 0.71, 95% CI 0.53–0.94, *p* = 0.017) (Table 2 and Appendix A).

#### 3.3.4. Electrolyte Disturbances

Compared with the patients without laxatives, the use of stimulant laxatives, docusate, or the combination of stimulant laxatives and docusate did not significantly increase or decrease the risk of hypokalemia, hypernatremia, and hypomagnesemia in both the logistic regression analysis and IPTW models (Table 2 and Appendix A).

### 3.4. Sensitivity Analyses

In the comparison between patients who used docusate and those who did not receive any laxatives, the E-value was 2.78 for in-hospital mortality. Therefore, the results of our study are moderately robust to potential unmeasured confounding, indicating that the observed associations between docusate use and in-hospital mortality were unlikely to be entirely explained by unmeasured confounders.

### 3.5. Subgroup Analysis

The subgroup analysis demonstrated that docusate reduced the in-hospital mortality in patients of age 65 years or older (OR: 0.46, 95% CI 0.28 to 0.72, *p* = 0.001) and those with severe clinical conditions, such as an APSIII score ≥ 40 (OR: 0.46, 95% CI 0.32 to 0.67, *p* < 0.001), CCI ≥ 3 (OR: 0.49, 95% CI 0.34 to 0.71, *p* < 0.001), or SOFA score ≥ 5 (OR: 0.48, 95% CI 0.33 to 0.69, *p* < 0.001) (Figure 3). Docusate was associated with a significantly lower mortality rate in patients without COPD (OR: 0.41, 95% CI 0.27 to 0.62, *p* < 0.001), without diabetes (OR: 0.44, 95% CI 0.27 to 0.7, *p* = 0.001), or without renal disease (OR: 0.42, 95% CI 0.27 to 0.64, *p* < 0.001), as seen in Figure 3.

## 4. Discussion

This was the first study to investigate the effects of prophylactic administration of stimulant laxatives and/or docusate on in-hospital mortality, ICU-free days and ventilator-free days at 28 days, and other clinical outcomes in critically ill, mechanically ventilated patients. Our findings suggest a possible association between the prophylactic use of docusate and a lower risk of in-hospital mortality and VAP, and an increase in ICU-free days at 28 days, without an increased risk of diarrhea, enterobacterial infections and electrolyte abnormalities.

Critically ill patients are susceptible to intestinal mucosal disturbances and increased intra-abdominal pressure due to impaired intestinal motility, facilitating intestinal bacterial translocation and promoting consequent enterobacterial infections [10]. Laxatives may alleviate bacterial and toxin accumulation by promoting the passage of intestinal contents [11]. Masri et al. [12] reported that prophylactic use of laxatives prevented constipation in ventilated ICU patients. Polyethylene glycol and lactulose promoted defecation in ICU patients [13]. Prophylaxis polyethylene glycol use resolved gastrointestinal tract paralysis faster than late intervention [14]. Daily use of lactulose induced defecation and reduced the SOFA scores in mechanically ventilated ICU patients [13]. However, lactulose increased the incidence of diarrhea. Senalin was more appropriate in ICU patients due to the similar efficacy and fewer complications than bisacodyl [15].

Stimulant laxatives are fast-acting and can induce defecation within 6–8 h after ingestion in a dose-dependent manner. High doses of stimulant laxatives stimulated sodium secretion into the colonic lumen, causing osmolyte transfer, leading to abdominal cramping and severe diarrhea [16]. The chronic use of stimulant laxatives was not recommended due to the adverse reactions [17,18]. Docusate, on the other hand, is a surfactant, which facilitates the emulsification of fats and oils in the stool. This emulsification is critical for softening the stool, particularly in critically ill patients, where bowel motility may be compromised due to sedation, immobility, or underlying conditions [19]. In general, docusate is a safer choice compared to stimulant laxatives.

Our study found that the use of docusate may be linked to a decrease in in-hospital mortality, whereas previous research did not find this effect with the use of other laxatives [12,13,20]. This difference could potentially be attributed to a lower prevalence of diarrhea and ventilator-associated pneumonia, as well as to more ICU-free days, in patients who received docusate compared to those who did not receive laxatives in our study. Diarrhea has been associated with increased mortality and a prolonged ICU stay in previous studies [21,22]. Ventilator-associated pneumonia was associated with increased mortality [23]. Furthermore, a longer duration of ICU stay has been identified as a strong risk factor for mortality [24]. These factors together may have contributed to the lower in-hospital mortality rate observed in patients who received docusate.

Our analysis indicated that the potential benefits of docusate appeared to be more pronounced in patients with greater clinical severity, such as those with an APSIII score ≥ 40, a Charlson Index score ≥ 3, or a SOFA score ≥ 5, particularly in individuals without COPD, diabetes, or renal disease. However, the current body of published research on the impact of docusate in critically ill patients is limited. While our findings suggest potential benefits, further studies are necessary to understand the mechanisms of the observed effects and to validate the findings. Additionally, factors such as complex surgical interventions, severe acute pathologies, or cardiovascular conditions, which may influence prolonged hospitalization or the use of mechanical ventilation, were not explored in this study. Future research should aim to address these limitations and provide a more comprehensive understanding of docusate’s role in critically ill populations.

Constipation predisposed ICU patients to respiratory infections due to exacerbated unaware aspiration of refluxed gastric contents precipitated by increased intra-abdominal pressure [25]. Previous studies have found that patients who had early defecation had fewer infections compared to those with late defecation [6]. This study found that prophylactic use of laxatives might not affect the risk of enterobacterial infection. It is also observed that laxatives might not be associated with an increased risk of C. difficile infection, which accords with previous research [26].

A major concern regarding the prophylactic use of laxatives in critically ill patients was the potential for adverse events and complications, such as diarrhea and electrolyte disorders. Most studies did not support the use of lactulose in ICU patients due to the increased risk of adverse reactions [20,27]. Hay et al. [28] found earlier prophylactic use of docusate-based regimens did not provide benefits compared to delayed laxative bowel intervention in mechanically ventilated ICU adults. Our study showed laxatives did not appear to substantially increase the incidence of diarrhea, even in patients who used both stimulant laxatives and docusate. Compared to non-laxatives, stimulant laxatives, docusate, or the combination of stimulant laxatives and docusate did not increase the risk of electrolytes disturbance.

### Strengths and Limitations

This was the first study to explore the impact of the prophylactic use of different types of laxatives on in-hospital mortality, ICU-free days at 28 days, ventilator-free days at 28 days, and other clinical outcomes in ventilated critically ill patients. To control the confounders and reduce the bias, the IPTW adjustment was utilized to enable these groups to be comparable. There were several limitations of this study. It is important to consider the limitations of this study when interpreting our findings. First, this study was a single-center study, which may limit the generalizability of our findings. Moreover, the bowel protocol in the ICU department at this center may differ from other centers in various countries. While the use of an open-source online database can provide high-quality data, it may introduce certain limitations due to variations in definitions, coding, protocols, and other management variables utilized over the extended period from 2008 to 2019 at a single medical institution. The strict screening criteria utilized in this study led to the exclusion of many patients. However, compared to other studies, the sample size was still relatively large in our study. The final sample size of 2129 patients, including 1245 patients taking no laxatives and 884 treated individuals divided into three subgroups, may have limited the statistical power to detect strong clinical end points, such as mortality. Third, the doses, frequencies, and durations of different laxatives were not consistent in this study. Finally, being a retrospective study, inherent limitations such as selection bias and potential unidentified confounding factors may have been present. However, the use of E-value analysis provided reassurance regarding the robustness of our results, indicating that the impact of unidentified variables on the observed associations was minimal. Multi-center randomized controlled trials are needed to identify an optimal prophylactic laxative regimen, dose and duration in ICU patients requiring mechanical ventilation.

## 5. Conclusions

Prophylactic use of docusate may be associated with a decreased risk of in-hospital mortality and VAP, as well as an increase in ICU-free days at 28 days. Notably, the use of prophylactic laxatives may not be associated with increased risk of diarrhea, enterobacterial infection, and electrolyte disturbance. Additional research is essential to validate these observations and determine the optimal dosage regimen for prophylactic laxative use in this specific population. Future studies should aim to clarify the potential benefits and risks associated with various laxative treatments in this patient group to inform clinical practice effectively.

## Figures and Tables

**Figure 1 diseases-12-00274-f001:**
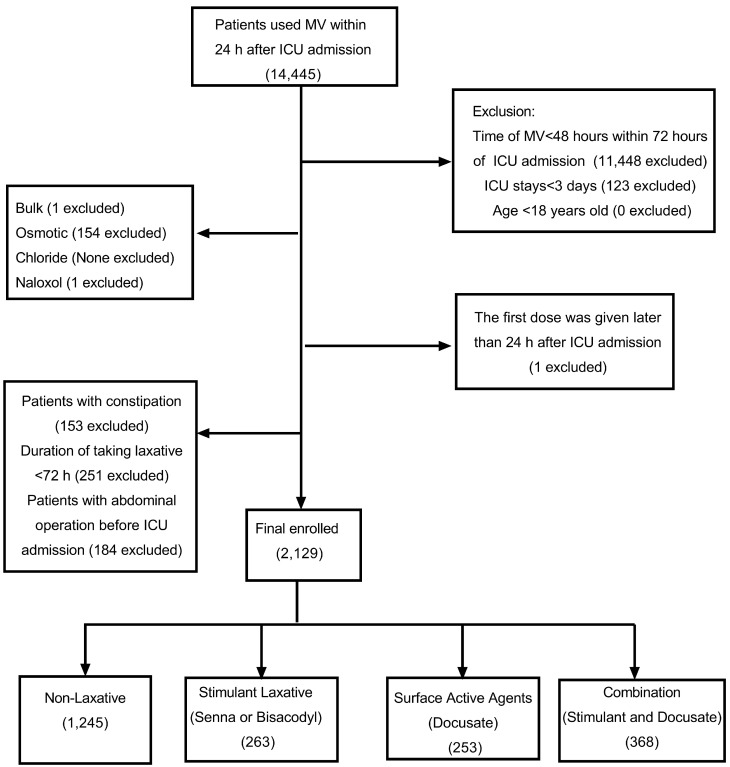
Flow diagram of the placement of patients in various groups.

**Figure 2 diseases-12-00274-f002:**
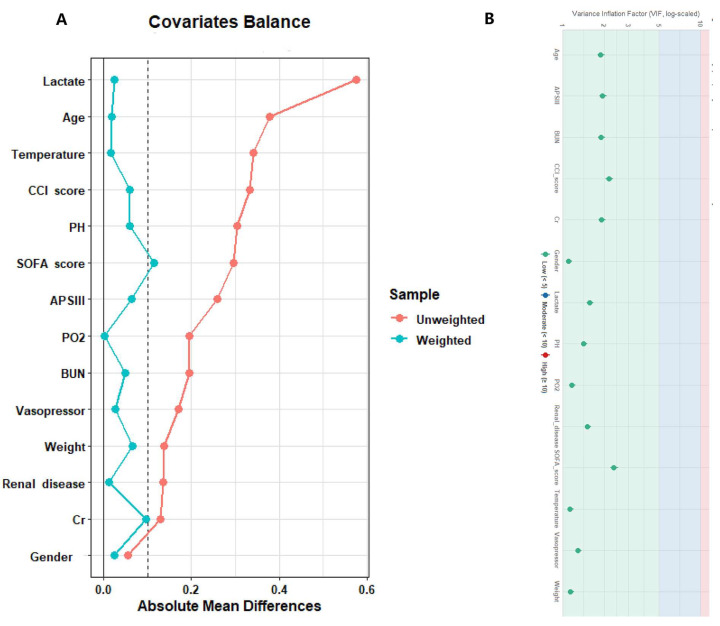
(**A**) Standardized mean difference (SMD) of the variables before and after weighting. The inverse probability treatment weighting (IPTW) significantly reduced the SMDs to less than 0.1, expect for the SOFA score. (**B**) Multicollinearity and variance inflation factor in the regression model. The variance inflation factor of each covariate not exceeding 3 meant no obvious multicollinearity requiring correction. Abbreviation: BUN, blood urea nitrogen; APSIII, Acute Physiology Score III; SOFA, Sequential Organ Failure Assessment; CCI, Charlson Comorbidity Index; SCr, serum creatine.

**Figure 3 diseases-12-00274-f003:**
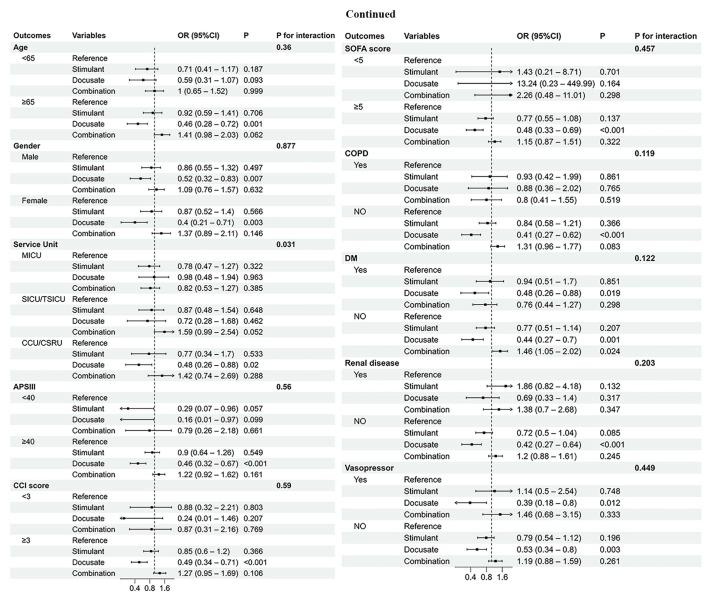
Subgroup analysis of the relationship between groups and in-hospital mortality. Abbreviation: APSIII, Acute Physiology Score III; SOFA, Sequential Organ Failure Assessment; CCI, Charlson Comorbidity Index; DM, diabetes; COPD, chronic obstructive pulmonary disease; OR, odds ratio; CI, confidence interval; MICU, medical intensive care unit; SICU, surgical intensive care unit; TSICU, trauma surgical intensive care unit; CCU, coronary care unit; CSRU, cardiac surgery recovery unit.

**Table 1 diseases-12-00274-t001:** Baseline characteristics of the original cohort.

	ICU Laxative Use				
	Non-Laxative (Reference)	Stimulants	Docusate	Combination	*p*-Value
Covariates	n = 1245	n = 263	n = 253	n = 368	
Age (year)	63.3 [50.4, 73.7]	68.9 [57.8, 78.2]	65.4 [53.3, 77.3]	63.6 [50.2, 75.1]	<0.001
Female (%)	521 (41.8)	118 (44.9)	96 (37.9)	151 (41.0)	0.453
Weight (kg)	79.2 [66.6, 98.2]	86.8 [74.0, 101.4]	79.2 [66.0, 97.0]	81.5 [67.9, 98.0]	0.234
Service unit (%)					<0.001
MICU	552 (44.3)	119 (45.2)	53 (20.9)	159 (43.2)	
SICU/TSICU	478 (38.4)	93 (35.4)	35 (13.8)	137 (37.2)	
CCU/CSRU	215 (17.3)	51 (19.4)	165 (65.2)	72 (19.6)	
APSIII	71.0 [49.0, 94.0]	70.0 [50.0, 89.0]	70.0 [51.8, 91.2]	75.0 [54.0, 97.0]	0.001
SOFA score	9.0 [7.0, 12.0]	11.0 [8.0, 13.0]	9.0 [7.0, 12.0]	10.0 [7.0, 13.0]	<0.001
CCI score	5.0 [3.0, 8.0]	6.0 [5.0, 8.0]	6.0 [4.0, 8.0]	5.0 [3.0, 7.0]	<0.001
DM (Yes, %)	371 (29.8)	73 (27.8)	97 (38.3)	111 (30.2)	0.036
CHF (Yes, %)	322 (25.9)	80 (30.4)	97 (38.3)	129 (35.1)	<0.001
Liver disease (Yes, %)	79 (6.3)	12 (4.6)	6 (2.4)	14 (3.8)	0.028
Renal disease (Yes, %)	195 (15.7)	39 (14.8)	69 (27.3)	61 (16.6)	<0.001
COPD (Yes, %)	209 (16.8)	57 (21.7)	41 (16.2)	76 (20.7)	0.114
Vasopressor (Yes, %)	232 (18.6)	33 (12.5)	69 (27.3)	37 (10.1)	<0.001
Opioid (Yes, %)	1141 (91.6)	239 (90.9)	241 (95.3)	336 (91.3)	0.213
Enteral nutrition (Yes, %)	277 (22.2)	92 (35.0)	46 (18.2)	152 (41.3)	<0.001
Temperature (C)	37.56 [37.0, 38.3]	37.2 [35.9, 37.8]	37.6 [36. 9, 38.5]	37.5 [35.9, 38.3]	<0.001
HR (bmp)	105 [69, 122]	90 [68, 114]	98 [67, 121]	109 [68, 128]	<0.001
MAP (mmHg)	69 [67, 116]	69 [49, 95]	69.0 [49.0, 115.0]	69.0 [49.0, 111.0]	0.009
WBC (K/uL)	14.4 [11.1, 19.6]	16.3 [11.2, 22.3]	13.6 [10.2, 18.2]	15.1 [10.7, 21.0]	0.008
Hb (g/L)	10.3 [8.7, 11.7]	8.5 [7.3, 10.5]	10.3 [8.4, 12.0]	9.6 [8.0, 11.7]	<0.001
Platelet (K/uL)	185.0 [130.0, 244.0]	123.0 [90.0, 178.0]	174.0 [123.8, 227.0]	156.0 [102.0, 216.0]	<0.001
pH	7.3 [7.2, 7.4]	7.3 [7.2, 7.4]	7.3 [7.2, 7.4]	7.3 [7.2, 7.4]	<0.001
pO2 (mmHg)	61.0 [43.0, 88.0]	70.0 [48.0, 89.0]	70.0 [48.8, 101.2]	65.0 [42.0, 90.0]	0.015
pCO2 (mmHg)	49.0 [42.0, 60.0]	50.0 [45.0, 58.0]	48.0 [40.0, 58.0]	48.0 [42.0, 57.0]	0.821
Potassium (mmol/L)	4.4 [3.8, 5.1]	4.5 [3.9, 5.0]	4.3 [3.6, 4.9]	4.4 [3.8, 5.1]	0.221
Sodium (mmol/L)	141.0 [139.0, 145.0]	141.0 [139.0, 144.0]	141.0 [139.0, 145.0]	141.0 [138.0, 144.0]	0.133
Magnesium (mg/dL)	1.7 [1.6, 1.9]	2.0 [1.8, 2.2]	1.8 [1.6, 1.9]	1.7 [1.5, 1.9]	<0.001
Chloride (mmol/L)	102.0 [99.0, 107.0]	104.0 [101.0, 107.0]	103.0 [99.0, 107.0]	103.0 [98.0, 107.0]	0.032
BUN (mg/dL)	21.0 [17.0, 26.0]	21.0 [20.0, 25.0]	21.0 [19.0, 26.0]	21.0 [17.0, 25.0]	0.005
SCr (mg/dL)	25.0 [16.0, 41.5]	22.0 [16.0, 33.0]	23.0 [15.0, 38.0]	25.0 [17.0, 42.0]	0.157
Lactic acid (mmol/L)	1.2 [0.8, 2.2]	1.3 [0.9, 1.9]	1.1 [0.8, 1.9]	1.3 [0.9, 2.2]	<0.001
Patient outcomes					
In-hospital mortality (Yes, %)	378 (30.4)	65 (24.7)	48 (19.0)	119 (32.3)	<0.001
ICU-free day (days)	19.7 [14.3, 22.6]	19.2 [14.0, 22.4]	19.4 [13.0, 22.3]	19.1 [13.7, 22.2]	0.151
VFD (days)	23.0 [20.0, 25.0]	23.0 [19.0, 25.0]	24.0 [21.0, 25.0]	23.0 [19.0, 25.0]	0.103
Diarrhea (Yes, %)	595 (47.8)	118 (44.9)	97 (38.3)	159 (43.2)	0.034
VAP (Yes, %)	202 (16.2)	48 (18.3)	46 (18.2)	65 (17.7)	0.756
Enterobacteria infection (Yes, %)	267 (21.4)	55 (20.9)	53 (20.9)	67 (18.2)	0.61
C. difficile infection (Yes, %)	36 (2.9)	7 (2.7)	6 (2.4)	10 (2.7)	0.972
Hypernatremia (Yes, %)	14 (1.1)	0 (0)	5 (2.0)	5 (1.4)	0.186
Hypokalemia (Yes, %)	249 (20.0)	41 (15.6)	51 (20.2)	66 (17.9)	0.354
Hypomagnesemia (Yes, %)	213 (17.1)	46(17.5)	46 (18.2)	55 (14.9)	0.706

Abbreviation: BUN, blood urea nitrogen; APSIII, Acute Physiology Score III; SOFA, Sequential Organ Failure Assessment; CCI, Charlson Comorbidity Index; DM, diabetes, CHF, congestive heart failure; COPD, chronic obstructive pulmonary disease; HR, heart rate; MAP, mean arterial pressure, WBC, white blood cell; Hb, hemoglobin; SCr, serum creatinine, MICU, medical intensive care unit; SICU, surgical intensive care unit; TSICU, trauma surgical intensive care unit; CCU, coronary care unit; CSRU, cardiac surgery recovery unit; VFD, ventilator-free days, VAP: ventilator-associated pneumonia. All the continuous variables are presented as the median (interquartile range).

**Table 2 diseases-12-00274-t002:** Analysis of the associations among laxatives and the clinical outcomes with the IPTW model.

Outcomes	Non-Laxative	Stimulant Laxatives		Docusate		Stimulants–Docusate	
		OR (95% CI)	*p*	OR (95% CI)	*p*	OR (95% CI)	*p*
**Primary outcome**							
In-hospital mortality	Ref	1.03 (0.76–1.41)	0.834	0.59 (0.42–0.83)	0.002	1.28 (0.98–1.67)	0.069
**Major secondary outcomes**							
ICU-free days *	Ref	0.95 (0.88–1.02)	0.137	0.89 (0.83–0.95)	<0.001	0.99 (0.93–1.05)	0.665
VFD *	Ref	0.96 (0.93–0.99)	0.022	1 (0.97–1.04)	0.8	1 (0.97–1.04)	0.623
**Other secondary outcomes**							
Diarrhea	Ref	1.01 (0.76–1.34)	0.952	0.71 (0.53–0.94)	0.017	0.94 (0.73–1.21)	0.635
VAP	Ref	0.97 (0.74–1.27)	0.82	0.62 (0.47–0.81)	0.001	0.85 (0.67–1.08)	0.182
Enterobacteria infection	Ref	1.06 (0.76–1.48)	0.713	1.03 (0.74–1.42)	0.874	0.9 (0.67–1.21)	0.478
C. difficile infection	Ref	0.75 (0.32–1.78)	0.515	0.69 (0.29–1.67)	0.414	0.8 (0.39–1.64)	0.545
Hypernatremia	Ref			1.56 (0.52–4.65)	0.429	0.96 (0.32–2.83)	0.936
Hypokalemia	Ref	0.75 (0.53–1.06)	0.101	1.04 (0.75–1.44)	0.819	0.78 (0.58–1.05)	0.108
Hypomagnesemia	Ref	1.01 (0.71–1.42)	0.972	1.15 (0.82–1.6)	0.425	0.73 (0.53–1.01)	0.056

* Continuous variables use β instead of OR. Abbreviations: Ref, reference; IPTW, inverse probability treatment weighting; OR, odds ratio; CI, confidence interval. VAP: ventilator-associated pneumonia.

## Data Availability

The data are available on the MIMIC-IV website at https://mimic.physionet.org/ (accessed on 1 October 2023). The datasets are available from the Massachusetts Institute of Technology (MIT) and Beth Israel Deaconess Medical Center (BIDMC) upon reasonable request according to the instructions for gaining access to MIMIC-IV. The data in this article can be reasonably requested by applying to the corresponding author.

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
