# Peer review of "The Effects of Prophylactic Laxative Use on Critically Ill Patients Requiring Mechanical Ventilation: A Retrospective Cohort Study"

_diseases, 2024, doi:10.3390/diseases12110274_

Round 1
Reviewer 1 Report
Comments and Suggestions for Authors
Interesting read and well written report. Suggest to add biochemical and physiological perspectives in the discussion to expand on the clinical understanding of the use of Docusate.
Author Response
Comment 1: Interesting read and well written report. Suggest to add biochemical and physiological perspectives in the discussion to expand on the clinical understanding of the use of Docusate.
Response 1:
Thank you for your valuable feedback and for recognizing the quality of our report. We appreciate your insightful suggestion to incorporate biochemical and physiological perspectives in the discussion to enhance the clinical understanding of Docusate use. We had revised the manuscript accordingly by expanding on these aspects in order to provide a more comprehensive view. [Line 292-295]
Reviewer 2 Report
Comments and Suggestions for Authors
This is a very interesting manuscript dealing with the role of stimulant laxatives and/or docusate on clinical outcomes in patients who require mechanical ventilation (MV).
My main concerns were addressed by the authors in the section Strengths & Limitations of the study. Minor comments are as follows:
- What was the length of use of the the laxative?
- Table 1. Hard to follow, please align the columns.
- Same for Table 2.
- Although mortality was almost twice as high in the combination group compared to individual treatments, why no differences were observed in the ICU-free days?
Author Response
Comments 1: What was the length of use of the laxative?
Response 1: Thank you for your question regarding the length of laxative use. In our study, we did not specifically collect the exact duration of laxative use for each patient due to the variability in hospitalization times, which is one of the inherent limitations of a retrospective study. However, we ensured that all patients received the laxative within the first 24 hours of hospitalization and continued its use for at least three days. We believe that this consistency in treatment protocol supports the validity of our results despite the lack of precise duration data.
Comments 2: Table 1. Hard to follow, please align the columns. Same for Table 2.
Response 2:
Thank you for your feedback on the tables. We understand that the formatting of Table 1 and Table 2 may have been unclear. We have revised the tables by aligning the columns to ensure they are easy to follow and more visually consistent.
Comments 3: Although mortality was almost twice as high in the combination group compared to individual treatments, why no differences were observed in the ICU-free days?
Response 3:
Thank you for your comments. Indeed, as shown in Table 1, docusate was associated with a lower in-hospital mortality rate compared to the combination group, yet no reduction in ICU-free days was observed. This phenomenon could be attributed to several factors. ICU patients often have multiple comorbidities, and their overall recovery may not rely solely on pharmacological interventions but also on surgeries, rehabilitation, and other non-drug treatments. These additional interventions may extend the length of hospital stay. Even if the primary disease is well controlled by the medications, other comorbidities still require treatment, which could prolong the hospitalization. While the medication might reduce mortality, it may not directly shorten the time needed to manage these comorbidities, as is evident from the fact that docusate did not reduce the duration of ventilator use in our study. Reducing in-hospital mortality and shortening the length of stay may be influenced by multiple factors. In some cases, a drug may effectively improve survival rates, but due to the time required for recovery or managing complications, the length of hospital stay may not correspondingly decrease.
Reviewer 3 Report
Comments and Suggestions for Authors
The topic is interesting but the study has many limitations.
It is in fact a single-center, analytical retrospective observational study, therefore with all the important limitations related to this design, in the quality and quantity of data available for analysis.
As is known, this type of study has the fundamental limit of the impossibility of selecting patients a posteriori and, therefore, the possibility that the result of the study (in both a specifically statistical and, consequently, clinical sense) is modulated by variables that the authors cannot control; modulation that could occur both in the direction of refuting the test hypothesis of the study and in the sense of leading to the conclusion that the test hypothesis of the study is true.
The authors try to overcome these limitations with a powerful statistical framework, which certainly "contains the damage" but cannot substantially overcome it, as can only be done by prospective, randomized and controlled studies on the subject.
In addition, the final sample size, i.e. 2129 patients, of which 1245 untreated and 884 treated and divided into three subgroups, appears inadequate to obtain a test power for strong clinical endpoints, such as mortality.
The use of an open source online database as a data source, although of high quality, may introduce a further limitation due to different standards of definition, coding, protocols and other management variables used, over a long period of time (2008-2019), at a medical institution different from that of those who perform the data analysis.
It is therefore necessary to completely revise paragraphs 4 and 5 (Discussion and Conclusion) (in particular, the following lines: 267-273, 292-296, 302-304, 310-312, 319-323, 342-348), reducing the emphasis on the results and toning down the statements on the related clinical implications, in order not to convey scientifically misleading messages.
Furthermore, the authors report a result worthy of further analysis, namely that in the subgroup analysis the presumed protective effect of docusate, found on strong clinical end-points such as in-hospital mortality and VAP, would be more pronounced in patients who are clinically more critical based on the severity scores used in intensive care, but who do not present the comorbidities that generally increase the risk of such patients, such as COPD, diabetes and renal disease.
It would therefore be interesting to know the causes of prolonged hospitalization and prolonged need for mechanical ventilation of these patients (complex surgical interventions? severe acute pathologies? etc.); the role played in the clinical severity or in the onset of complications and in mortality by the presence of cardiovascular pathologies; the causes of death of deceased patients. All these data could contribute to a more in-depth interpretation of the results.
Author Response
Comments 1: The topic is interesting, but the study has many limitations.
It is in fact a single-center, analytical retrospective observational study, therefore with all the important limitations related to this design, in the quality and quantity of data available for analysis. As is known, this type of study has the fundamental limit of the impossibility of selecting patients a posteriori and, therefore, the possibility that the result of the study (in both a specifically statistical and, consequently, clinical sense) is modulated by variables that the authors cannot control; modulation that could occur both in the direction of refuting the test hypothesis of the study and in the sense of leading to the conclusion that the test hypothesis of the study is true. The authors try to overcome these limitations with a powerful statistical framework, which certainly "contains the damage" but cannot substantially overcome it, as can only be done by prospective, randomized and controlled studies on the subject.
Response 1: We acknowledge that our study, being a single-center, retrospective observational analysis, has inherent limitations, particularly concerning the quality and quantity of available data. We have revised the Discussion section to describe these limitations and their potential impact on our findings, particularly regarding the robustness of our conclusions. We also emphasize the necessity for prospective, randomized controlled trials to address these issues definitively (Line 341-363).
Comments 2: In addition, the final sample size, i.e. 2129 patients, of which 1245 untreated and 884 treated and divided into three subgroups, appears inadequate to obtain a test power for strong clinical endpoints, such as mortality.
Response 2:
We appreciate your concerns regarding the sample size. We have clarified in the revised manuscript that while our sample of 2,129 patients, divided into treated and untreated groups, provides meaningful insights, it may limit the power to detect strong clinical endpoints like mortality (Line 354-356).
Comments 3: The use of an open-source online database as a data source, although of high quality, may introduce a further limitation due to different standards of definition, coding, protocols and other management variables used, over a long period of time (2008-2019), at a medical institution different from that of those who perform the data analysis.
Response 3:
We recognize the challenges associated with using an open-source online database for data analysis, particularly regarding variations in definitions, coding standards, and protocols over time. We have added a statement on this limitation, emphasizing the importance of considering these factors when interpreting our results (Line 349-352).
Comments 4: It is therefore necessary to completely revise paragraphs 4 and 5 (Discussion and Conclusion) (in particular, the following lines: 267-273, 292-296, 302-304, 310-312, 319-323, 342-348), reducing the emphasis on the results and toning down the statements on the related clinical implications, in order not to convey scientifically misleading messages.
Response 4: We have carefully revised these lines of the Discussion and Conclusion sections to tone down statements about our results and to ensure our statements regarding clinical implications are presented with appropriate caution (Line 271, 297-299, 308-309,327-328).
Comments 5: Furthermore, the authors report a result worthy of further analysis, namely that in the subgroup analysis the presumed protective effect of docusate, found on strong clinical end-points such as in-hospital mortality and VAP, would be more pronounced in patients who are clinically more critical based on the severity scores used in intensive care, but who do not present the comorbidities that generally increase the risk of such patients, such as COPD, diabetes and renal disease. It would therefore be interesting to know the causes of prolonged hospitalization and prolonged need for mechanical ventilation of these patients (complex surgical interventions? severe acute pathologies? etc.); the role played in the clinical severity or in the onset of complications and in mortality by the presence of cardiovascular pathologies; the causes of death of deceased patients. All these data could contribute to a more in-depth interpretation of the results.
Response 5: Thank you for your valuable feedback and insightful suggestions. We agree that the subgroup analysis indicating a more pronounced protective effect of docusate in clinically severe patients, such as those with APSIII scores ≥40, Charlson Index scores ≥3, or SOFA scores ≥5, warrants further exploration. Additionally, the observation that this effect appeared to be stronger in patients without comorbidities such as COPD, diabetes, or renal disease is intriguing. However, the current body of research on this topic is limited, and we were unable to gather additional evidence to explain why docusate appears to be more effective in critically ill patients. Your suggestions, particularly the exploration of factors such as prolonged hospitalization, the need for mechanical ventilation, complex surgical interventions, acute pathologies, cardiovascular conditions, and causes of death, provide excellent directions for future research. We have incorporated these insights into our Discussion and emphasized the need for further studies to validate and expand upon these findings (Lines 314-322).
Reviewer 4 Report
Comments and Suggestions for Authors
The article "The effects of prophylactic laxative use on critically ill patients requiring mechanical ventilation" is a significant retrospective study that aimed to examine the impact of prophylactic use of laxatives in critically ill patients requiring mechanical ventilation. The authors analyzed the effectiveness of different types of laxatives, such as sodium docusate and stimulants, in terms of their effect on in-hospital mortality and the occurrence of ventilator-associated pneumonia (VAP).
Strengths of the Article:
- Clarity: The article's structure is well-organized and clear, which helps in understanding the methods used and the results obtained.
- Statistical Analysis: The use of the IPTW (Inverse Probability Treatment Weighting) method minimized bias in evaluating the effects of the treatment.
- Findings: Prophylactic use of docusate was associated with reduced in-hospital mortality and a lower risk of VAP, which is valuable for clinical practice.
Limitations of the Article:
- Study Limitations: The study is based on data from a single medical facility, which may limit the generalizability of the results to other patient populations.
- Retrospective Nature: The retrospective analysis, despite attempts to minimize errors, might be affected by various confounding factors.
In conclusion, the article provides important insights into the prophylactic use of laxatives in critically ill patients. However, further studies are necessary to confirm the findings and optimize therapeutic protocols.
Author Response
Comments 1: Study Limitations: The study is based on data from a single medical facility, which may limit the generalizability of the results to other patient populations.
Response 1: Thank you for your feedback. We acknowledge that conducting our study within a single medical facility may limit the generalizability of our results to broader patient populations (Line 346). To address this concern, we have revised the Discussion section to emphasize the need for multicenter studies that could validate our findings (Line 361).
Comments 2: Retrospective Nature: The retrospective analysis, despite attempts to minimize errors, might be affected by various confounding factors.
Response 2: We appreciate your comment. We acknowledged that retrospective study design might be affected by confounders, although we tried different ways to minimize potential confounding factors. Additionally, we have highlighted the importance of considering these limitations when interpreting our findings (Line 341-363).
Comments 3: In conclusion, the article provides important insights into the prophylactic use of laxatives in critically ill patients. However, further studies are necessary to confirm the findings and optimize therapeutic protocols.
Response 3: We agree that further studies are essential to confirm our findings and optimize therapeutic protocols related to the prophylactic use of laxatives in critically ill patients. We have made this point more prominent in the revised Discussion and Conclusions sections to encourage future research in this area (Line 368-372).
Round 2
Reviewer 3 Report
Comments and Suggestions for Authors
The new version of the Discussion and Conclusions paragraphs highlights the limitations of the study more correctly than the previous version and consequently scales down the scientific value of the reported results and emphasizes more the need for further studies and data on the topic.
A rereading of the text is the only further suggestion, only to correct some minor spelling errors.